# Solvent-free bottom-up patterning of zeolitic imidazolate frameworks

Yurun Miao [1,7], Dennis T. Lee [1,7], Matheus Dorneles de Mello [2,3], Mueed Ahmad[2,4], Mohammed K. Abdel-Rahman[5], Patrick M. Eckhert [5], J. Anibal Boscoboinik [2,4], D. Howard Fairbrother[5] & Michael Tsapatsis [1,6✉]

Patterning metal-organic frameworks (MOFs) at submicrometer scale is a crucial yet challenging task for their integration in miniaturized devices. Here we report an electron beam (e-beam) assisted, bottom-up approach for patterning of two MOFs, zeolitic imidazolate frameworks (ZIF), ZIF-8 and ZIF-67. A mild pretreatment of metal oxide precursors with linker vapor leads to the sensitization of the oxide surface to e-beam irradiation, effectively inhibiting subsequent conversion of the oxide to ZIFs in irradiated areas, while ZIF growth in non-irradiated areas is not affected. Well-resolved patterns with features down to the scale of 100 nm can be achieved. This developer-free, all-vapor phase technique will facilitate the incorporation of MOFs in micro- and nanofabrication processes.

[1] Department of Chemical and Biomolecular Engineering & Institute for NanoBioTechnology, Johns Hopkins University, Baltimore, MD, USA. [2] Center for Functional Nanomaterials, Brookhaven National Laboratory, Upton, NY, USA. [3] Catalysis Center for Energy Innovation, University of Delaware, Newark, DE, USA. [4] Department of Materials Science and Chemical Engineering, Stony Brook University, Stony Brook, NY, USA. [5] Department of Chemistry, Johns Hopkins University, Baltimore, MD, USA. [6] Applied Physics Laboratory, Johns Hopkins University, Laurel, MD, USA. [7] These authors contributed equally: Yurun Miao, Dennis T. Lee. ✉email: tsapatsis@jhu.edu

MOFs are isoreticular coordination networks assembled from metal ions and organic linkers[1–3]. They hold promise in applications including gas storage and separation[4–9], catalysis[10], drug delivery[11], and energy conversion[12], due to their modular nature allowing for excellent tunability in the structural and chemical properties[13]. Hence, significant work has been dedicated to controlling the composition, dimensions, and positioning of MOF crystals and films by manipulating various synthetic parameters as well as introducing innovative approaches[14,15]. There is a long-standing interest in the development of patterning process for porous materials[16–20], and in particular, MOF patterns with sub-micrometer scale precision offer unique advantages in their potential use in electronic and optical devices[21–24]. Recently, it has been demonstrated that amorphization of ZIFs, under X-ray[25] and e-beam[25–28] irradiations enables a selective removal of the irradiated or non-irradiated regions of the ZIFs in a liquid phase developing step. Deep X-ray lithography was also utilized to pattern ZIF films by selectively crosslinking a sol-gel bottom layer[29]. Despite these promising developments in top-down MOF patterning, there is no demonstration of bottom-up approaches using X-ray or e-beam, which have the potential to reach higher resolution than light-based systems with UV irradiation[30] or with infrared laser writing[31]. Moreover, top-down patterning methods[25–27,29] rely on irradiation-induced solubility change and subsequent removal of materials by dissolution. Yet, solvent-free approaches for patterning are currently at the forefront of technological needs due to their great potential in improving wafer processing efficiency and patterning quality at reduced material and energy cost[32]. Although solvent-free MOF deposition steps have been incorporated in lift-off patterning[33,34], fully solvent-free bottom-up patterning of MOFs will further facilitate their application in microfabrication processes.

Here, we report a developer-free, all-vapor, e-beam-induced area-selective bottom-up approach for the patterning of ZIFs. It achieves high precision (down to ca. 200 nm ZIF-8 line width and 100 nm gap in between) in ZIF position and size, outperforming other MOF bottom-up patterning approaches[35–40] and broadening the applicability of e-beam MOF patterning.

## Results

The processing technique employs the vapor-phase conversion of oxide films (ZnO and $CoO_x$) to crystalline ZIFs using sublimated vapors of 2-methylimidazole (2mIm)[21,33]. The key element introduced here is the pretreatment of the oxide layer with 2mIm at a relatively low temperature (ca. 50 °C) compared to the temperature used for ZIF crystallization (ca. 100 °C). This mild pretreatment aims to form an adsorbed 2mIm layer that sensitizes the oxide surface to e-beam treatment while avoiding ZIF crystallization. We hypothesize that the 2mIm-sensitized oxide film can be altered by e-beam treatment to render it less reactive towards ZIF crystallization, allowing ZIF formation preferentially in the non-irradiated areas.

For the demonstration of the patterning process, we focus on ZIF-8, a prototypical ZIF (Fig. 1a). The area-selective patterning starts from a precursor layer of 15-nm-thick zinc oxide deposited on a silicon substrate via atomic layer deposition (ALD) and consists of three consecutive steps: (i) the ZnO layer is sensitized by exposure to 2mIm vapor at relatively low temperature, 50 °C for 1 h; (ii) the regions on the sensitized ZnO surface are irradiated using a direct-write focused e-beam (2 keV, 20 mC cm$^{-2}$); (iii) the sensitized and e-beam-patterned ZnO film is treated with 2mIm vapor at 100–120 °C for 15–120 min to complete the patterning process by converting ZnO to ZIF-8 in the non-irradiated area, while ZIF-8 growth in the e-beam irradiated area is inhibited.

As illustrated in Fig. 1b–d, well-defined ZIF-8 patterns with a variety of dimensions and shapes can be obtained. The patterns on silicon wafer substrates were characterized by atomic force microscopy (AFM) and scanning electron microscopy (SEM). AFM images show straight line patterns of polycrystalline ZIF-8 "ridges" in the non-irradiated areas and smooth, ZIF-8-free "trenches" coinciding with the e-beam irradiated area (Fig. 1b). The gap between ZIF-8 lines is as low as ca. 100 nm and the center-to-center spacing down to 300 nm (200 nm of ZIF-8 line and 100 nm of gap) were achieved. The thinnest lines obtained are about 200 nm in width and consist of small intergrown crystals.

Applying "hole" (disc and square) patterns in the e-beam writing process leads to the crystallization of ZIF-8 in isolated regions, where poly- or single-crystalline ZIF-8 particles are arranged in a hexagonal or square lattice (Fig. 1c, d). The grain structure appears to be dependent on the size of non-irradiated "holes". ZIF-8 particles grown from a region smaller than 150 nm in size are mostly single crystalline (Fig. 1d), possibly due to a lower probability of multiple nucleation events in a single domain. Since the patterned deposit in this bottom-up approach is the outcome of crystal growth, the edge roughness of the deposit, the fidelity by which it fills the desired (non-irradiated areas), and the degree of spilling over to the irradiated areas, depend on the ability to control nucleation and growth in the non-irradiated areas. To improve ZIF pattern fidelity to the pattern created by e-beam irradiation, potential approaches include controlling the preferential orientation and the polycrystallinity (grain size) of the deposit. For example, we anticipate that if the grain size of the ZIF deposit can be reduced to a few unit cells (ca. 3 nm), resolution in the 10 nm range can be achieved.

To allow for crystallographic studies, ZIF patterning on 50 nm-thick silicon nitride windows was attempted (Fig. 2a). The obtained patterns were characterized by transmission electron microscopy (TEM) (Fig. 2b and Supplementary Fig. 1) and AFM (Fig. 2d), clearly displaying well-resolved polycrystalline structures. The slightly bright lining at the edge of the polycrystalline region shown in the TEM images is ascribed to the depletion of local ZnO precursor contributing to ZIF formation. In contrast, in the surrounding dark area, dense ZnO is well preserved. The crystallinity of the ZIF-8 deposit is confirmed by the ring patterns obtained with selected area electron diffraction (SAED) (Fig. 2c). ZnO remains smooth in an e-beam exposed area, and no diffraction is observed, indicating that the reactivity of ZnO to 2mIm can be suppressed entirely by the sequential combination of sensitization and e-beam irradiation.

A series of e-beam irradiations and 2mIm vapor treatments were investigated to assess the range for ZIF-8 patterning. A ZnO film sensitized with 2mIm was irradiated using an array of four squares (2 µm × 2 µm) with different electron doses (1, 5, 10, and 20 mC cm$^{-2}$, respectively) and subsequently exposed to 2mIm vapor at 100 and 130 °C for 15 min (Fig. 3a, b). At each temperature, the crystallization behavior was progressively altered by increased e-beam irradiation. The squares exposed to 1 mC cm$^{-2}$ doses show slightly larger grains than those in the non-irradiated region. Increased irradiation leads to fewer and smaller crystals in each square. Upon reaching a threshold dose (10 mC cm$^{-2}$ for 100 °C 2mIm vapor treatment and 20 mC cm$^{-2}$ for 130 °C 2mIm vapor treatment), ZIF-8 formation is inhibited, and the surface within the irradiated squares is smooth and free of ZIF-8 crystals.

A cross-section of a region encompassing both ZIF-8 and ZIF-8-free adjacent areas (Fig. 3c) was prepared by focused ion beam (FIB) and examined by TEM (Fig. 3d, e) to elucidate the structure after the 2mIm vapor treatment. In the square irradiated with 20 mC cm$^{-2}$, the ZnO film remains intact, confirming that its conversion to ZIF-8 is entirely suppressed by the e-beam

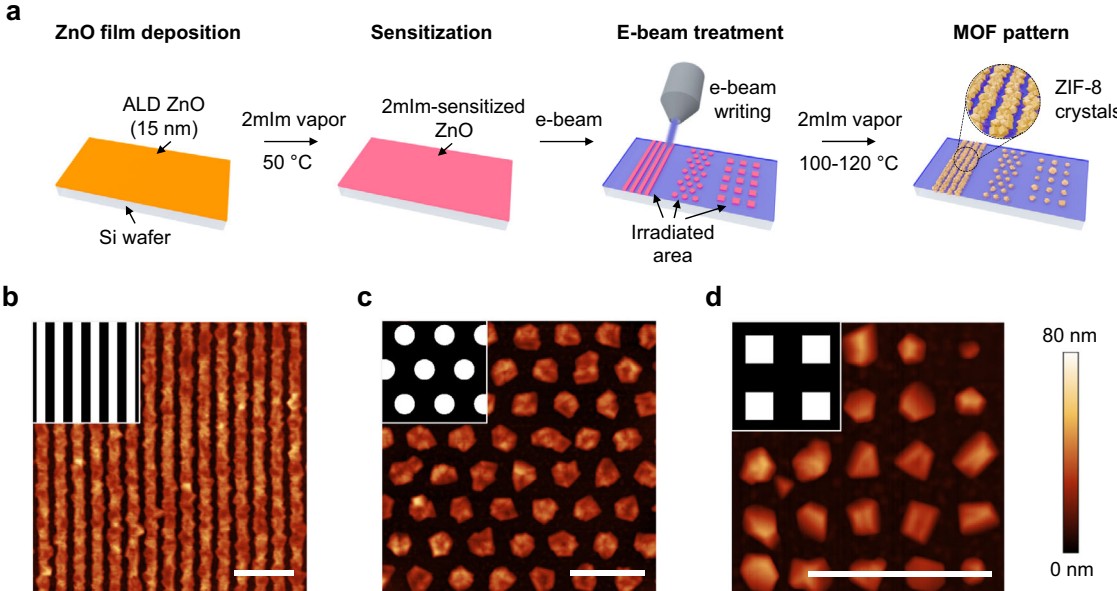

**Fig. 1 ZIF-8 patterns prepared by area-selective deposition on silicon wafer substrates. a** schematic illustration of the e-beam assisted area-selective deposition of ZIF-8, showing patterns with different designs. **b** AFM image of a line pattern with 150 nm width and 150 nm gap. **c, d** AFM images of ZIF-8 prepared by e-beam irradiation using a pattern of **c** 280 nm discs (corresponding to non-irradiated areas) in a hexagonal lattice and **d** 150-nm squares (non-irradiated areas) arranged in a square lattice, respectively. Center-to-center distances between neighboring discs and squares are 560 and 300 nm, respectively. Irradiated and non-irradiated areas are marked in black and white, respectively, in the top left inserts in **b, c**, and **d**. All patterns in this figure were prepared by sensitizing ZnO with 2mIm at 50 °C for 1 h, followed by e-beam patterning (2 keV, 20 mC cm$^{-2}$) and 2mIm vapor treatment at 100 °C for 15 min. Scale bars are 1 μm.

irradiation. In contrast, the ZnO in the non-irradiated area is mostly consumed after the vapor treatment. In agreement with previous reports[33,41], a thin unconverted layer of ZnO remains. It is located at the substrate-ZnO interface consistent with the proposed conversion of ZnO to ZIF-8 starting from the top of the film and propagating to the substrate-ZnO interface[41]. The presence of this thin unconverted layer could be beneficial for ensuring good adhesion of the ZIF-8 deposit to the substrate[33]. The ZIF-8 and ZIF-8-free areas are also clearly distinguished in SEM-EDS (Supplementary Fig. 2), corresponding to the C/N-rich and C/N-deficient areas respectively.

The duration of the ligand vapor treatment for ZIF-8 growth was extended beyond 15 min to investigate how the deposits evolve with time (Supplementary Figure 3). After 1 h at 100 °C, while the thickness of ZIF-8 in the non-irradiated area increases from 50 to 200 nm due to continued growth, the squares irradiated with 10 and 20 mC cm$^{-2}$ are still smooth without significant nucleation. However, after 2 h at 120 °C, ZIF-8 is present in all the irradiated squares, and the pattern cannot be distinguished from the background.

The adsorption of imidazole or its derivatives to carbon steel and copper has been studied in the context of enhancing anti-corrosion efficiency[42,43]. For ZnO, it was reported that a slight increase in film thickness (ca. 3 nm) was observed by in-situ ellipsometry at the initial stage (ca. 10 min) of the 2mIm vapor treatment of a ZnO thin film. This thickness increase was attributed to the sorption of 2mIm[41]. We systematically characterized samples obtained at various processing steps along the route to the selective ZIF growth to identify the species present on the ZnO film surface after the sensitization step and determine any changes in their chemical, morphological, and crystalline properties. In these experiments, the entire area of a sensitized ZnO film was exposed to the output of an electron gun operating at 2 kV and characterized by AFM and grazing incidence X-ray diffraction (GIXD) (Fig. 4a, b). GIXD shows no evidence for

crystallization throughout the sensitization and e-beam treatment process (Fig. 4a, traces for ALD ZnO, ZnO+s, ZnO+s+e), while no change in roughness is found from AFM images both after sensitization (ZnO+s) and after e-beam irradiation (ZnO+s+e) (Fig. 4b), confirming the absence of ZIF-8. The irradiated sample was further treated with 2mIm vapor at 120 °C for 1 h (ZnO+s +e+2mIm). After the 2mIm vapor treatment, the surface remains flat with only a few scattered small dots (<10 nm in height). On the other hand, non-sensitized but e-beam irradiated ZnO (ZnO+e+2mIm) shows uniform coverage of ZIF-8 growth on the substrate after 2mIm vapor treatment, and so does ZnO with sensitization but without e-beam irradiation (ZnO +s+2mIm), clearly substantiating the necessity of both sensitization and e-beam treatment to selectively inhibit the ZnO conversion to ZIF-8. Infrared reflection absorption spectroscopy (IRRAS) results for the samples (Supplementary Fig. 4) also show a good agreement with the conclusions drawn from GIXD and AFM, especially in that the characteristic vibrational modes (1500–1200 cm$^{-1}$ wavenumber) of the 2mIm in ZIF-8 are mostly absent in the "ZnO+s+e+2mIm" sample, while they are present in "ZnO+e+2mIm" and "ZnO+s+2mIm" samples.

X-ray photoelectron spectroscopy (XPS) was also performed to assess chemical changes on the surface of the samples (Fig. 4c and Supplementary Figs. 5–7). After the sensitization at 50 °C for 1 h (ZnO+s), a peak appears at 398.8 eV with a shoulder at 400.6 eV in N 1s XPS (Fig. 4c). The peak at 398.8 eV corresponds to N in the unreacted imidazole molecule or N bound to Zn, while the peak at 400.6 eV is assigned to protonated N (N–H). The different peak intensities could be a consequence of the adsorbate geo-metry with the bound N lying closest to the surface[44]. After e-beam irradiation (ZnO+s+e), the overall intensity of the N 1s region decreases, and the N–H shoulder cannot be detected, possibly due to electron-induced deprotonation, desorption, and crosslinking of the adsorbed 2mIm[45,46]. The passivation is likely a result of the e-beam induced crosslinking that transforms the

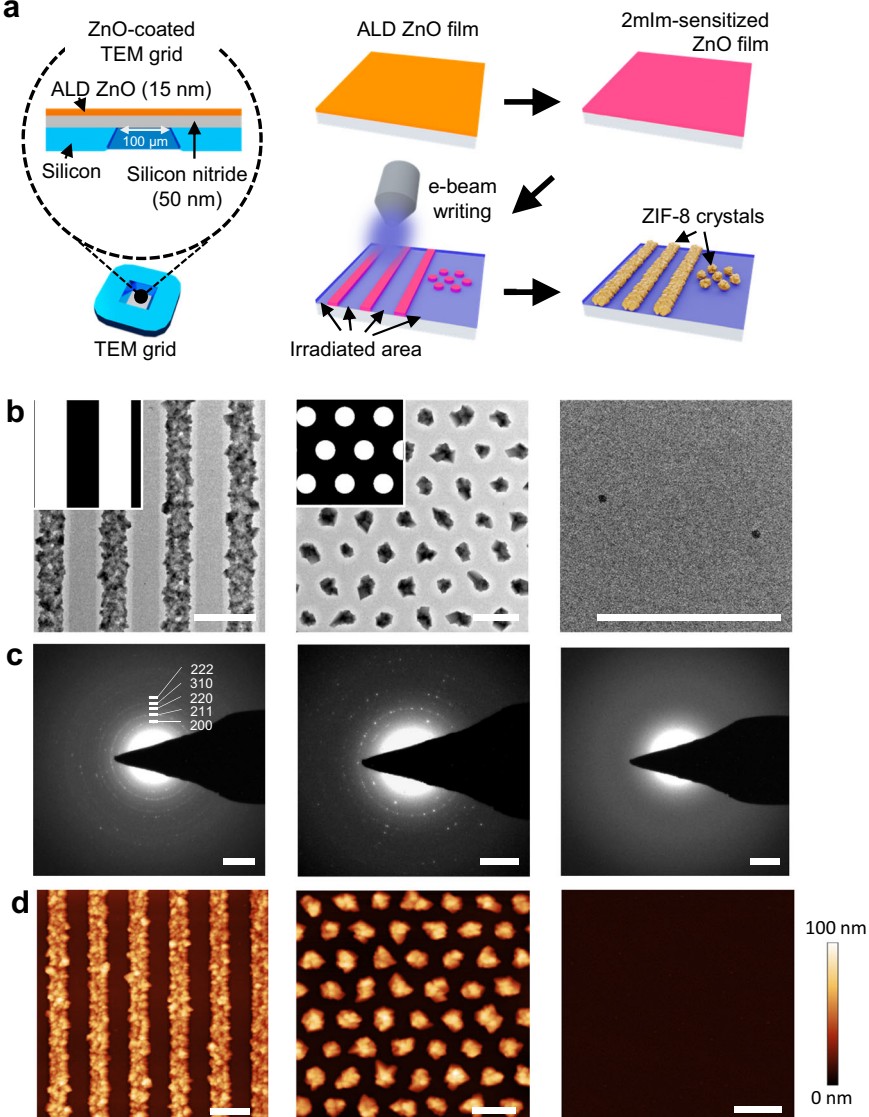

**Fig. 2 ZIF-8 patterns on silicon nitride windows. a** schematic illustration of the patterning process on silicon nitride substrate. **b** TEM images of line (500 nm width and 500 nm spacing) and dot (400 nm diameter and 400 nm spacing) patterns and a completely irradiated (non-growth) area, respectively. Irradiated and non-irradiated areas are marked in black and white, respectively, in the top left insets of line and dot patterns in **b**. **c** SAED patterns of the imaged areas in **b**; diffraction rings corresponding to ZIF-8 are indexed for the line pattern. **d** AFM height images of different areas on the silicon nitride window corresponding to the TEM images in **b**. Scale bars are 1 μm in **b** and **d**, and 1 nm$^{-1}$ in **c**.

adsorbed 2mIm molecules into an oligomerized or polymeric coating, which inhibits the conversion of ZnO to ZIF-8 in the irradiated area, while the non-irradiated ZnO surface remains reactive during the 2mIm vapor treatment. This crosslinking and partial desorption hypothesis is consistent with XPS data, which shows the disappearance of the N–H component of the N 1s region after e-beam exposure of the sensitized sample and a decrease in overall N 1s intensity. The areal dose used in this study (20 mC cm$^{-2}$ at 2 kV) is relatively high compared to typical resists used in e-beam lithography (<1 mC cm$^{-2}$)[47]. Future research may focus on exploring similar adsorptive species with higher e-beam sensitivity to reduce the write time and improve the processing efficiency of this approach. The peak at 400.6 eV reemerges after the vapor treatment (ZnO+s+e + 2mIm), probably because of 2mIm molecules adsorbed during the vapor treatment. In contrast, only the major peak at 398.7 eV, which is characteristic for ZIF-8[48], is observed in the samples lacking sensitization (ZnO+e + 2mIm) or e-beam irradiation (ZnO

+s + 2mIm). This agrees well with the assignment of the low BE peak to N atoms forming part of the framework (bound to Zn). Further analysis of C 1s spectra (Supplementary Fig. 5) supports the hypothesis of crosslinking. (ZnO+s) shows three components at 284.8, 285.5, and 287.2 eV assigned to C–C/C–H, C–N, and C–O bonds, respectively. The latter indicates some C–O–Zn linkages upon sensitization with 2mIm. Exposure to the e-beam (ZnO+s + e) leads to a broadening of the component at 285.5 eV assigned to C–N bonds, which suggests the formation of species of different chemical environments[49].

In the current approach, the passivated ZnO remains in the areas that do not convert to ZIF. For certain applications, the presence of ZnO may not be undesirable. For instance, when patterned MOFs are used as diffraction grating sensors, the diffraction intensity depends on the refractive index difference in different regions of the pattern[25,50,51]. The non-MOF part of the pattern can be either empty or filled with metal oxide[50] as long as there is a spatial contrast in the refractive index.

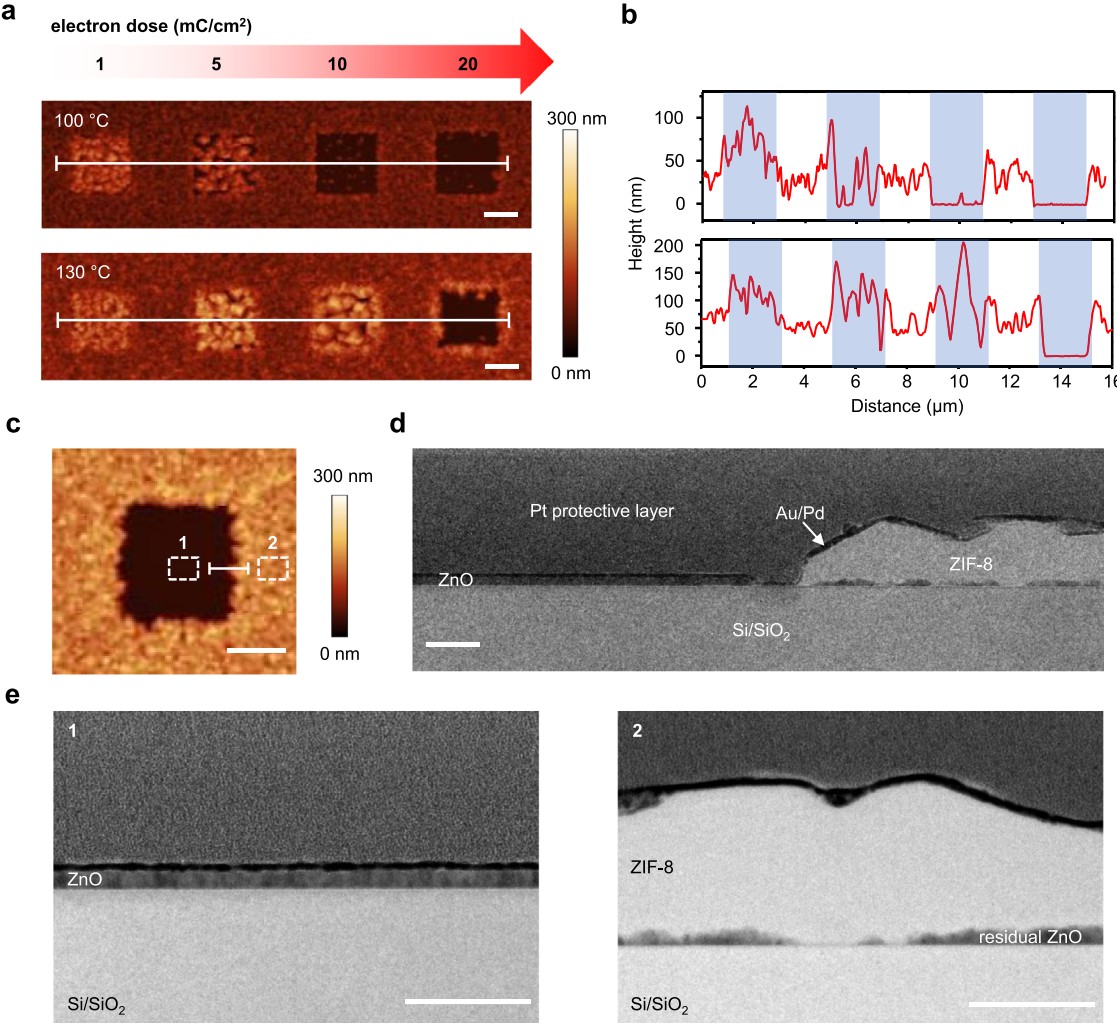

**Fig. 3 ZIF-8 growth under various reaction conditions. a** AFM height mapping of samples after vapor treatment at different temperatures. ZnO wafers are sensitized by 2mIm at 50 °C for 1 h, and each of the four squares is irradiated with an electron dose of 1, 5, 10 and 20 mC cm$^{-2}$ at 2 kV, respectively, before the vapor treatment with 2mIm at 100 or 130 °C for 15 min. **b** line-scan profile across the marked trace for each sample in **a**. Shaded regions correspond to areas irradiated by e-beam. **c** AFM image of a ZIF-8-free square in a ZIF-8 film prepared by sensitizing a ZnO wafer with 2mIm at 50 °C for 1 h, followed by e-beam irradiation (2 kV, 20 mC cm$^{-2}$) of a 2 μm × 2 μm square and vapor treatment with 2mIm at 120 °C for 15 min. **d, e** TEM images of the cross-section prepared by FIB corresponding to the white line (**d**) and marked areas 1 and 2 (**e**) in **c**. Scale bars are 1 μm in **a** and **c**, and 100 nm in **d** and **e**.

A sensitization approach was introduced and demonstrated to enable e-beam-based bottom-up patterning of ZIF-8. It relies on finding a window of processing conditions that allow linker incorporation in the precursor oxide film without causing ZIF-8 nucleation. The subsequent writing with an e-beam passivates the exposed areas and determines where ZIF-8 would nucleate and grow. This method should be in principle applicable to other ZIFs and MOFs. In preliminary experiments, we demonstrate feasibility for patterning of ZIF-67, a Co ZIF, by following similar treatment steps (but with adjustments in duration) as for ZIF-8 while using cobalt oxide (CoO$_x$) as the substrate. First, an ALD CoO$_x$ film was sensitized with 2mIm at 50 °C for 2 h. AFM images show that the surface roughness of CoO$_x$ remains essentially unchanged after sensitization (Fig. 5a). After e-beam irradiation (2 kV, 20 mC cm$^{-2}$) and subsequent vapor treatment with 2mIm at 120 °C for 2 h (Fig. 5b), the irradiated area remains smooth as opposed to the significantly roughened non-irradiated area due to the growth of ZIF-67. Consistently, SEM-EDS shows a relatively lower content of C and N in the irradiated area due to the lack of CoO$_x$ conversion to ZIF-67 (Supplementary Fig. 8).

## Methods

**Materials.** 2-methylimidazole (2mIm, 99%) was purchased from Sigma Aldrich. Diethylzinc (DEZ, 95% purity), tetrakis(dimethylamino)zirconium(IV) (TDMAZ, 99% purity), and bis(*N-t*-butyl-*N′*-ethylpropanimidamidato)cobalt(II) (Co(AMD)$_2$, min. 98%) were purchased from STREM chemicals Inc. Homemade Milli-Q DI-water (H$_2$O) was used. Silicon wafers were purchased from UniversityWafer Inc., and silicon nitride supports (50 nm silicon nitride film on a 200 μm silicon frame with nine viewing windows, each 0.1 × 0.1 mm) for transmission electron microscopy (TEM) were purchased from Ted Pella.

**Atomic layer deposition (ALD) on planar substrates.** Several pieces of Si wafer (1 cm × 1 cm size) with thin native oxide (~2 nm) (Si/SiO$_2$) and silicon nitride supports were loaded into the ALD reactor (Savannah S200, Veeco Instruments Inc.). Prior to the ALD processes, the silicon nitride supports were treated with oxygen plasma (29.6 W, 400 mTorr) for 10 min to improve the surface reactivity with ALD precursors using a plasma cleaner (Harrick Plasma).

The inorganic ZnO thin film was deposited with H$_2$O/DEZ precursor pulses separated by Ar purge in between. The sequence time (in second) of "H$_2$O pulse/Ar purge/DEZ pulse/Ar purge" was "0.015/5/0.015/5" for ZnO deposition. The ZnO-coated Si wafers were prepared with 100 cycles of ALD ZnO to the Si/SiO$_2$ at 125 °C under ~0.270 Torr. Around 15 nm of ZnO film on the Si/SiO$_2$ was confirmed from an FS-1 Multi-Wavelength Ellipsometer System.

In addition, CoO$_x$ film deposition was performed separately. To provide enough vapor pressure of Co(AMD)$_2$ during the film deposition, a stainless-steel cylinder container with the metal precursor was heated to 100 °C, and the vapor was doubly

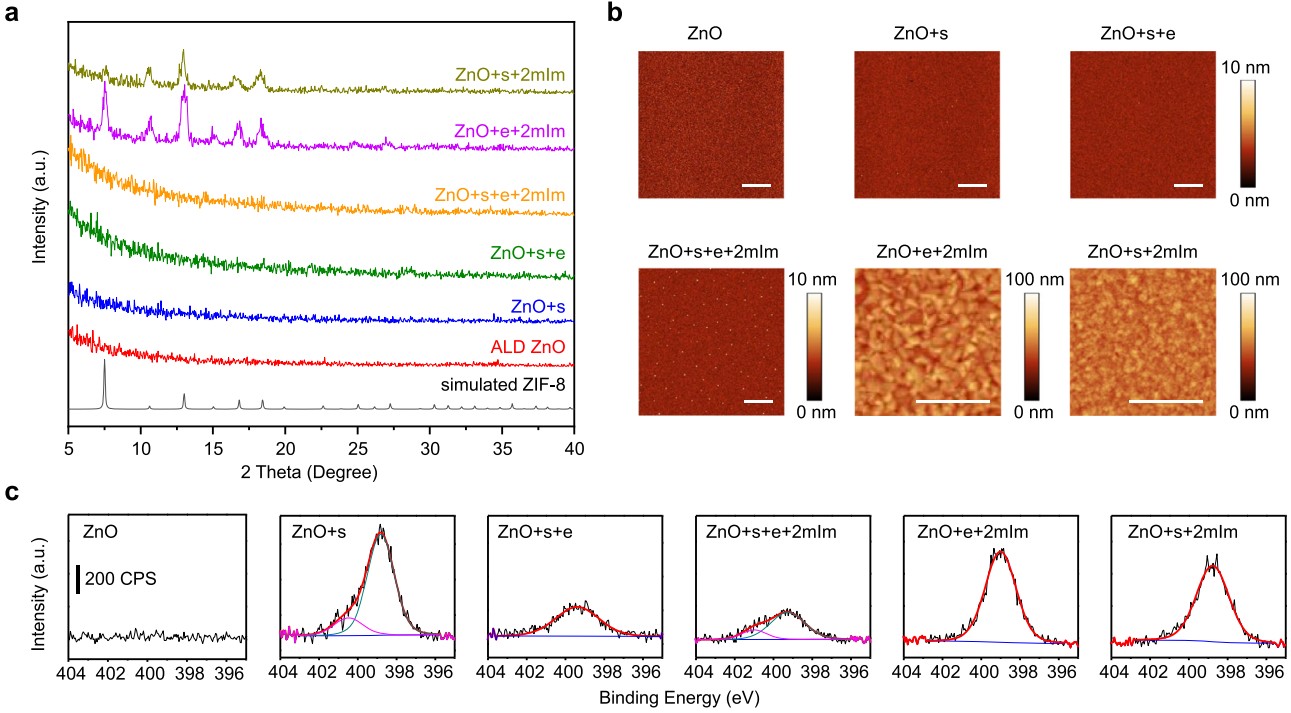

**Fig. 4 Characterization of sensitized and e-beam irradiated ZnO samples. a, b** GIXD and AFM of a ZnO wafer before and after consecutive treatment of sensitization, e-beam irradiation, and 2mIm treatment. Samples are denoted to indicate the processing steps used, s: sensitization with 2mIm at 50 °C for 1 h; e: e-beam irradiation at 2 kV, 20 mC cm$^{-2}$; 2mIm: vapor treatment with 2mIm at 120 °C for 15 min. GIXD patterns were collected at a 0.15° incidence angle. **c**. N 1$s$ XPS of samples corresponding to **a**. Scale bars are 1 μm.

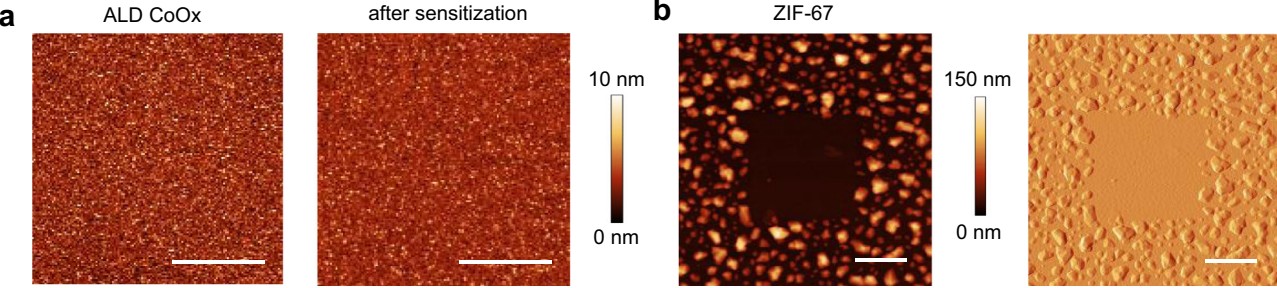

**Fig. 5 Selective deposition of ZIF-67 from CoO$_x$. a** AFM images of an ALD CoO$_x$ film before and after sensitization with 2mIm at 50 °C for 2 h. **b** AFM height (left) and amplitude error (right) images of a ZIF-free area on a ZIF-67 film, prepared by e-beam irradiation (2 kV, 20 mC cm$^{-2}$) of a sensitized CoO$_x$ sample in a 2 μm × 2 μm square, followed by vapor treatment with 2mIm at 120 °C for 2 h. Scale bars are 1 μm.

pulsed to the reactor. Around 20 nm of ZrO$_2$ film as an adhesion layer was deposited to Si/SiO$_2$ substrates prior to the CoO$_x$ ALD processes to enhance film uniformity. The CoO$_x$-coated samples were prepared with 150 cycles of ALD CoO$_x$ to the ZrO$_2$/(Si/SiO$_2$) at 150 °C under ~0.570 Torr. The sequence time (in seconds) of "H$_2$O pulse/Ar purge/(Co(AMD)$_2$ pulse/Ar purge)×2" was "0.015/5/(0.5/20)×2" for CoO$_x$ deposition. Around 9 nm of CoO$_x$ film on the ZrO$_2$/(Si/SiO$_2$) was confirmed from an FS-1 Multi-Wavelength Ellipsometer System.

**Sensitization of metal oxide thin film.** The prepared metal oxide-coated samples were placed on a stainless-steel mesh holder and transferred to a cylindrical quartz reactor system. First, 0.2 g of 2mIm solid was loaded at the bottom of the reactor. The reactor was then connected to a vacuum pump via a manual valve, and the whole set-up was placed in the oven. The system was flushed with an Ar gas (100 sccm) at room temperature under a dynamic vacuum (~50 mbar) for 15 min. After that, the Ar flushing was stopped, and the reactor was evacuated for 5 min until the system pressure stabilizes at the base pressure (~10 mbar). Then, the valve to the vacuum pump was closed, followed by flowing the Ar to the system until the system pressure reaches around 1 bar. The reactor was heated at 50 °C for 1 h and then cooled down to room temperature.

**E-beam patterning.** E-beam exposure experiments were performed on a ThermoFisher Helios G4 UC Focused Ion Dual Beam microscope. Patterning was

performed on the sensitized samples at 2 kV acceleration voltage with 100 pA probe current. All patterns were exposed at 1 μs dwell time and 2.2 nm pitch size, while the pass (scan) number was varied in each exposure to obtain the desired electron dose.

**ZIF film growth.** The e-beam patterned samples were placed on a stainless-steel mesh holder and transferred to a quartz reactor system. The system was flushed with an Ar gas and evacuated under a vacuum in the same way as the metal oxide sensitization process. After that, the valve to the vacuum pump was closed, followed by heating the reactor at 120 °C for 2 h under static vacuum. After the reaction is over, the reactor was cooled down to room temperature. The produced films were stored in a desiccator before being used for further characterizations.

**E-beam irradiation.** The entire surface of wafers was irradiated in a home-built UHV system equipped with an electron gun (ELG-2, Kimball Physics) mounted on a UHV chamber. Wafer samples were loaded into the UHV chamber and evacuated overnight before irradiation. Chamber pressure was kept below 1 × 10$^{-7}$ mTorr during irradiation. Beam parameters were controlled by a separate power supply unit (EGPS-1022, Kimball Physics). The emission current was set at 17.2 μA, corresponding to an electron flux of 1.6 μA cm$^{-2}$ at 30 mm spot size and 100 mm working distance.

**Characterization**. Grazing Incidence X-ray Diffraction (GIXD) analysis of the thin film materials was performed using a Rigaku Smartlab (Instrument at BNL). Films were scanned at 40 kV and 45 mA using Cu Kα radiation ($\lambda = 1.54$ Å) and a step size of $2\theta = 0.04°$ ($2°$ min$^{-1}$) over a $2\theta$ range of 5–40°. Incidence angles varied from 0.1 to 0.20°. AFM images were collected on a Bruker Multimode 8 with a Si tip at a scan rate of 1 Hz and 256 lines/scan under tapping mode. Around 2.5 nm Au/Pd coating was deposited on all the Si wafer-based samples using a Leica EM ACE600 sputter for obtaining scanning electron microscopy (SEM) images. The SEM images and energy-dispersive X-ray spectroscopy (EDS) data were collected using a ThermoFisher Helios G4 UC Focused Ion Dual Beam microscope. TEM images and ED patterns were obtained on a ThermoFisher TF30 TEM operating at 300 kV. X-ray photoelectron spectroscopy (XPS) experiments were performed using a system equipped with a SPECS PHOIBOS NAP 150 hemispherical analyzer and a monochromatic Al Kα X-ray source. The spectra were acquired under UHV conditions (base pressure of $2 \times 10^{-9}$ mbar) and 298 K on sample areas smaller than 300 μm × 300 μm. The spectra were calibrated using C 1$s$ main peak at 284.8 eV as reference. Infrared reflection absorption spectroscopy (IRRAS) was performed in a Bruker Vertex 80 V spectrometer at grazing incidence (8°) under UHV (base pressure of $2 \times 10^{-9}$ mbar) and a mercury–cadmium–telluride (MCT) detector located in an external chamber. The spectra were collected for wavelengths in the range of 800–4000 cm$^{-1}$ with 500 scans and a 4 cm$^{-1}$ resolution. In this work, the $s$-polarized spectrum was collected as background and subtracted from the $p$-polarized spectrum for all experiments.

## Data availability

The data that support the findings of this study are available in the article or its Supplementary File, or available from the corresponding author on request.

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

## Acknowledgements

This material is based upon work supported by the U.S. Department of Energy, Office of Science, Office of Basic Energy Sciences, Division of Chemical Sciences, Geosciences and Biosciences under Award DE-SC0021212 and Award DE-SC0021304. XPS data collection and analysis was partially supported by the Catalysis Center for Energy Innovation, an Energy Frontier Research Center funded by the U.S. Department of Energy, Office of Science, Office of Basic Energy Sciences under Award No. DE-SC0001004. Surface characterization carried out in part at Center for Functional Nanomaterials at Brookhaven National Laboratory, supported by the U.S. Department of Energy, Office of Basic Energy Sciences, under Contract No. DE-SC0012704. M.T. and D.H.F. acknowledge partial support from a 2019 JHU Discovery Award (Design of Interfaces between Porous and Non-Porous Materials for Energy Applications).

## Author contributions

M.T. conceived and directed the project. Patterning experiments were designed by M.T., Y.M., and D.T.L. Experiments and data analysis on MOF patterning was performed by Y.M. and D.T.L. GIXD and XPS experiments were performed by M.D.d.M. and M.A., and analyzed with input from J.A.B. and D.H.F. IRRAS experiments were directed by J.A.B. E-beam experiments on entire wafers were performed by M.K.A.-R, P.M.E., and Y.M. under D.H.F. direction. The manuscript was written mainly by Y.M. and D.T.L. with contributions by M.T. and input from all authors.

## Competing interests

The authors declare no competing interests.
