## [Peer Review File · Nature Communications]

Solvent-Free Bottom-up Patterning of Zeolitic Imidazolate FrameworksREVIEWER COMMENTS

Reviewer #1 (Remarks to the Author):

In general, this is a nice study, offering though not so much insights. At some places, the study could have been worked out better (see also comments below). Nevertheless, the overall quality of the work is good, and of interest to the community.

The working principle behind the concept is not very well explored. What is the fate of the imidazole used in the 'sensitization step'? Likely it is cross-linked? Detailed XPS and infrared analysis (see also comment below on the IRRAS data) should enable the authors to formulate a well-founded hypothesis.

In addition, the ZnO (likely covered with cross-linked imidazole residue) remains present in between the MOF patterns. How would this method be used in a fabrication flow? I would like to invite the authors to share their vision on this point.

The experiment performed in Fig. 3c-e is very nice!

The authors state "there is no demonstration of irradiation-based bottom-up approaches". However, there are several reported studies in which light irradiation was used to trigger the nucleation of MOF materials, for instance: *Angewandte Chemie International Edition*, 2014, 53, 5561-5565 and *Chemical Communications*, 2017, 53, 5275-5278.

The line edge roughness is quite bad and seems to be determined by crystal facet formation. This effect is especially apparent in Fig. 1 c and d where the formed patterns deviate significantly from the irradiated pattern. Likely, this drawback is inherent to the presented 'bottom-up' approach. A comment should be included on this topic as line edge roughness is a key performance parameter in lithography. Also, it would be enlightening if the authors would propose a strategy to mitigate this effect.

How can the porosity of the MOF material patterned on the surface be verified?

The infrared characterization is not very convincing. The spectra shown in Fig. S4 are of poor quality and hardly show any convincing features. The quality of these spectra should be improved and the features of interest should be indicated and, if possible, referenced to literature. It is not clear if the 'IRRAS' principle was in fact used as the substrate used for these measurements was not metal-coated. Also, the vertical axis of Fig. S4 is confusingly labeled 'Transmittance'.

The required areal dose is high, especially compared with typical resists used in e-beam lithography. A high dose requirement puts a severe cap on the area that can be patterned (in a reasonable amount of time). It would be appropriate to include a comment on this as a perspective for future research.

Reviewer #2 (Remarks to the Author):

This paper reports a highly innovative bottom-up method for patterning metal organic frameworks on the substrate. In particular, this work demonstrates the patterning of two zeolitic imidazolate frameworks (ZIF-8 and ZIF-67) with well controlled structures. A mild treatment of metal oxide precursors with organic linker vapor is discovered to lead to the sensitization of the oxide surface, which is key to successful selective growth of ZIFs into different designed patterns. The solvent-free synthesis makes this patterning technique environmentally friendly and easy to integrate within wafer processing. This work represents an exciting breakthrough in the field. The experiments are carefully

designed and conducted, and demonstrate the sensitization and selective growth mechanisms. The manuscript is of very high quality. Therefore, I recommend that the manuscript be accepted for publication after a minor revision as suggested below.

In the patterning process, the precision of down to ca. 200 nm ZIF-8 line width and 100 nm gap in between in ZIF position and size is achieved. Can the authors explain what limits this precision? Please also comment on whether it is possible to further reduce line width and gap in the future study.

Reviewer #3 (Remarks to the Author):

The current contribution focus on a solvent-free bottom-up patterning approach for ZIF-8, but also generalized to other MOF compounds. Its basic approach for formation of the MOF material itself is based on the work by the Ameloot group in <http://dx.doi.org/10.1038/NMAT4509> which is also mentioned as ref. 36 in this work. What the current paper adds to this is to create a passivation layer on top of the oxide layer using an e-beam approach. By this, they encapsulate the layer and slow down the rate of nucleation of the ZIF-8 material upon exposure to the 2-methylimidazole. When done with the correct exposure times a patterned layer will be formed.

Coatings of MOF type of materials, and patterning of such, is an emerging field that is still lacking a practical application. When such is identified, the possibility to pattern MOFs will undoubtedly be important. As such, the content is novel and deserves attention. However, the current paper has some shortcomings that makes me hesitant to recommend it for Nature Communications. I will try to elaborate.

Bottom-up patterning of MOF materials was also included in the original paper by Ameloot [ref. 36]. See its figure 5a and the description in the text that goes with it. Rather than to passivate parts of the oxide material, they remove those parts that that should not be developed into a MOF material. The approach is also described as bottom-up approach in [Metal organic framework top-down and bottom-up patterning techniques <http://dx.doi.org/10.1039/D0DT02207A>]. The process can be completely dry depending on how the ZnO material is patterned.

In the approach suggested by the authors, the unconverted layer with a carbon passivation layer will remain in between their patterns. This is hardly mentioned in the paper, and neither of how this will affect the properties of the resulting device/structure.

In addition, similar approaches are also described using X-ray or e-beam lithography in the papers:

Top-down patterning of Zeolitic Imidazolate Framework composite thin films by deep X-ray lithography <http://dx.doi.org/10.1039/C2CC33292B>

Direct X-ray and electron-beam lithography of halogenated zeolitic imidazolate frameworks <https://doi.org/10.1038/s41563-020-00827-x>

The three latter works should be mentioned and dealt with in the current paper.

Additional thoughts:

Page 5 line 123: The authors show that the amount of ZIF-8 formed is linearly dependent on the exposure time. 15 min = 50 nm and 60 min = 200 nm. The way I interpret this is that the conversion process is not complete in the first case, and may still not be so after 1 h. Primarily: When will it saturate? Where is the excess ZnO stored before conversion, and how does this affect the properties?

Response to Reviewers for

Solvent-Free Bottom-up Patterning of Zeolitic Imidazolate Frameworks

*Changes to the manuscript are highlighted in yellow

Reviewer #1 (Remarks to the Author):

[COMMENT] In general, this is a nice study, offering though not so much insights. At some places, the study could have been worked out better (see also comments below). Nevertheless, the overall quality of the work is good, and of interest to the community.

[RESPONSE] We thank the reviewer for the thorough and constructive review.

[COMMENT] The working principle behind the concept is not very well explored. What is the fate of the imidazole used in the ‘sensitization step’? Likely it is cross-linked? Detailed XPS and infrared analysis (see also comment below on the IRRAS data) should enable the authors to formulate a well-founded hypothesis.

[RESPONSE] We agree that 2mIm is probably crosslinked in the e-beam treatment. As described in the discussion regarding Fig. 4c, N1s XPS shows the disappearance of the N-H bond after e-beam irradiation to the sensitized surface. We updated the discussion with the following text (on Page 6):

“X-ray photoelectron spectroscopy (XPS) was also performed to assess chemical changes on the surface of the samples (Fig. 4c and S5-7). After the sensitization at 50 °C for 1 h (ZnO+s), a peak appears at 398.8 eV with a shoulder at 400.6 eV in N 1s XPS (Fig. 4c). The peak at 398.8 eV corresponds to N in the unreacted imidazole molecule or N bound to Zn, while the peak at 400.6 eV is assigned to protonated N (N-H). The different peak intensities could be a consequence of the adsorbate geometry with the bound N lying closest to the surface.[44] After e-beam irradiation (ZnO+s+e), the overall intensity of the N 1s region decreases and the N-H shoulder cannot be detected, possibly due to electron-induced deprotonation, desorption, and crosslinking of the adsorbed 2mIm.[45,46] The passivation is likely a result of the e-beam induced crosslinking that transforms the adsorbed 2mIm molecules into an oligomerized or polymeric coating, which inhibits the conversion of ZnO to ZIF-8 in the irradiated area, while the non-irradiated ZnO surface remains reactive during the 2mIm vapor treatment. This crosslinking and partial desorption hypothesis is consistent with XPS data, which shows the disappearance of the N-H component of the N 1s region after e-beam exposure of the sensitized sample and a decrease in overall N 1s intensity. The areal dose used in this study (20 mC/cm² at 2 kV) is relatively high compared to typical resists used in e-beam lithography (<1 mC/cm²).[47] Future research may focus on exploring similar adsorptive species with higher e-beam sensitivity to reduce the write time and improve the processing efficiency of this approach. The peak at 400.6 eV reemerges after the vapor treatment (ZnO+s+e+2mIm), probably because of 2mIm molecules adsorbed during the vapor treatment. In contrast, only the major peak at 398.7 eV, which is characteristic for ZIF-8,[48] is observed in the samples lacking sensitization (ZnO+e+2mIm) or e-beam irradiation (ZnO+s+2mIm). This agrees well with the assignment of the low BE peak to N atoms forming part of the framework (bound to Zn). Further analysis of C 1s spectra (Fig. S5) supports the hypothesis of crosslinking. (ZnO+s) shows three components at 284.8, 285.5 and 287.2 eV assigned to C-C/C-H, C-N, and C-O bonds, respectively. The latter indicates some C-O-Zn linkages upon sensitization with 2mIm. Exposure to the e-beam (ZnO+s+e)

leads to a broadening of the component at 285.5 eV assigned to C-N bonds, which suggests the formation of species of different chemical environments.[49]”

In order to address the problem with a low intensity of the IRRAS signal, we prepared new films on gold-coated Si wafers, and repeated the experiments on these films. Given the selection rules in IRRAS, when a metal support is used, the signal intensity was greatly enhanced, aiding in a more reliable assignment of the vibrational modes. Changes have been made to Figure S4 and its caption accordingly. The spectra of (ZnO+s+2mIm) and (ZnO+e+2mIm) now show very clear characteristic bands of ZIF-8. However, it is difficult to identify the chemical species in films (ZnO+s) and (ZnO+s+e), which do not result in ZIF-8 formation, due to two main contributing factors: (1) since the ZnO layer is not consumed to form ZIF-8, this 15 nm oxide effectively inhibits most of the IRRAS signal enhancement yielded by gold in the substrate. Note that ZnO does not have any vibrational modes in the range of frequencies we have access to ($900\text{-}4000\text{ cm}^{-1}$), and only the hypothesized crosslinked passivating layer would have vibrational modes we could observe. However, the thin nature of this layer, and loss of signal enhancement for these samples that have a relatively thick unreacted oxide layer on the gold surface, prevent these modes from being observed.

[COMMENT] In addition, the ZnO (likely covered with cross-linked imidazole residue) remains present in between the MOF patterns. How would this method be used in a fabrication flow? I would like to invite the authors to share their vision on this point.

[RESPONSE] We add the following discussion to the text (on Page 6):

“In the current approach, the passivated ZnO remains in the areas that do not convert to ZIF. For certain applications, the presence of ZnO may not be undesirable. For instance, when patterned MOFs are used as diffraction grating sensors, the diffraction intensity depends on the refractive index difference in different regions of the pattern.[25,50,51] The non-MOF part of the pattern can be either empty or filled with metal oxide[50] as long as there is a spatial contrast in refractive index.”

[COMMENT] The experiment performed in Fig. 3c-e is very nice!

[RESPONSE] We thank the reviewer for recognizing our efforts.

[COMMENT] The authors state “there is no demonstration of irradiation-based bottom-up approaches”. However, there are several reported studies in which light irradiation was used to trigger the nucleation of MOF materials, for instance: Angewandte Chemie International Edition, 2014, 53, 5561-5565 and Chemical Communications, 2017, 53, 5275-5278.

[RESPONSE] The following text has been now updated (on Page 1) to be more accurate regarding comparisons with prior work:

“Despite these promising developments in top-down MOF patterning, there is no demonstration of bottom-up approaches using X-ray or e-beam, which have the potential to reach higher resolution than light-based systems with UV irradiation[30] or with infrared laser writing.[31]”

[COMMENT] The line edge roughness is quite bad and seems to be determined by crystal facet formation. This effect is especially apparent in Fig. 1 c and d where the formed patterns deviate significantly from the irradiated pattern. Likely, this drawback is inherent to the presented ‘bottom-up’ approach. A comment should be included on this topic as line edge roughness is a key performance parameter in lithography. Also, it would be enlightening if the authors would propose a strategy to mitigate this effect.

[RESPONSE] The following text has been added to the discussion regarding Fig. 1 (on Page 4).

“Since the patterned deposit in this bottom-up approach is the outcome of crystal growth, the edge roughness of the deposit, the fidelity by which it fills the desired (non-irradiated areas) and the degree of spilling over to the irradiated areas, depend on the ability to control nucleation and growth in the non-irradiated areas. To improve ZIF pattern fidelity to the pattern created by e-beam irradiation, potential approaches include controlling the preferential orientation and the polycrystallinity (grain size) of the deposit. For example, we anticipate that if the grain size of the ZIF deposit can be reduced to few unit cells (ca. 3 nm), resolution in the 10 nm range can be achieved.”

[COMMENT] How can the porosity of the MOF material patterned on the surface be verified?

[RESPONSE] As demonstrated by SAED patterns in Fig. 2c, the ZIF-8 components in our patterns are crystalline. Thus it is expected that their porosity is similar to that of other crystalline ZIF-8 reported. However, due to the small size of the patterned area in this study, we do not attempt to verify the porosity directly. We hope to address this issue in future work using larger area patterns.

[COMMENT] The infrared characterization is not very convincing. The spectra shown in Fig. S4 are of poor quality and hardly show any convincing features. The quality of these spectra should be improved and the features of interest should be indicated and, if possible, referenced to literature. It is not clear if the ‘IRRAS’ principle was in fact used as the substrate used for these measurements was not metal-coated.

[RESPONSE] We apologize for the poor quality of IR spectra. While IRRAS can be done on non-metallic supports, the signal is weak, and the vibrational modes require a more cumbersome interpretation (Chem. Soc. Rev., 2017,46, 1875-1932). Therefore, we prepared new films on gold-coated Si wafers and repeated the experiments on these films. Changes have been made to Figure S4 and its caption accordingly, as detailed in response to comment #1 above.

[COMMENT] Also, the vertical axis of Fig. S4 is confusingly labeled ‘Transmittance’.

[RESPONSE] We apologize for the mistake. The label of the vertical axis should be “Absorbance” and has been corrected.

[COMMENT] The required areal dose is high, especially compared with typical resists used in e-beam lithography. A high dose requirement puts a severe cap on the area that can be patterned (in a reasonable amount of time). It would be appropriate to include a comment on this as a perspective for future research.

[RESPONSE] The following text has been added to the discussion (on Page 6).

“The areal dose used in this study (20 mC/cm^2 at 2 kV) is relatively high compared to typical resists used in e-beam lithography ($<1 \text{ mC/cm}^2$).^[47] Future research may focus on exploring similar adsorptive species with higher e-beam sensitivity to reduce the write time and improve the processing efficiency of this approach.”

Reviewer #2 (Remarks to the Author):

[COMMENT] This paper reports a highly innovative bottom-up method for patterning metal organic frameworks on the substrate. In particular, this work demonstrates the patterning of two

zeolitic imidazolate frameworks (ZIF-8 and ZIF-67) with well controlled structures. A mild treatment of metal oxide precursors with organic linker vapor is discovered to lead to the sensitization of the oxide surface, which is key to successful selective growth of ZIFs into different designed patterns. The solvent-free synthesis makes this patterning technique environmentally friendly and easy to integrate within wafer processing. This work represents an exciting breakthrough in the field. The experiments are carefully designed and conducted, and demonstrate the sensitization and selective growth mechanisms. The manuscript is of very high quality. Therefore, I recommend that the manuscript be accepted for publication after a minor revision as suggested below.

[RESPONSE] We thank the reviewer for the thorough and constructive review.

[COMMENT] In the patterning process, the precision of down to ca. 200 nm ZIF-8 line width and 100 nm gap in between in ZIF position and size is achieved. Can the authors explain what limits this precision? Please also comment on whether it is possible to further reduce line width and gap in the future study.

[RESPONSE] We thank the reviewer for the recognition of our work. The following text has been added to the discussion regarding Fig. 1 (on Page 4).

“Since the patterned deposit in this bottom-up approach is the outcome of crystal growth, the edge roughness of the deposit, the fidelity by which it fills the desired (non-irradiated areas) and the degree of spilling over to the irradiated areas, depend on the ability to control nucleation and growth in the non-irradiated areas. To improve ZIF pattern fidelity to the pattern created by e-beam irradiation, potential approaches include controlling the preferential orientation and the polycrystallinity (grain size) of the deposit. For example, we anticipate that if the grain size of the ZIF deposit can be reduced to few unit cells of ZIF-8, resolution in the 10 nm range can be achieved.”

Reviewer #3 (Remarks to the Author):

[COMMENT] The current contribution focus on a solvent-free bottom-up patterning approach for ZIF-8, but also generalized to other MOF compounds. Its basic approach for formation of the MOF material itself is based on the work by the Ameloot group in <http://dx.doi.org/10.1038/NMAT4509> which is also mentioned as ref. 36 in this work. What the current paper adds to this is to create a passivation layer on top of the oxide layer using an e-beam approach. By this, they encapsulate the layer and slow down the rate of nucleation of the ZIF-8 material upon exposure to the 2-methylimidazole. When done with the correct exposure times a patterned layer will be formed.

Coatings of MOF type of materials, and patterning of such, is an emerging field that is still lacking a practical application. When such is identified, the possibility to pattern MOFs will undoubtedly be important. As such, the content is novel and deserves attention. However, the current paper has some shortcomings that makes me hesitant to recommend it for Nature Communications. I will try to elaborate.

[RESPONSE] We thank the reviewer for the thorough and constructive review.

[COMMENT] Bottom-up patterning of MOF materials was also included in the original paper by Ameloot [ref. 36]. See its figure 5a and the description in the text that goes with it. Rather than to

passivate parts of the oxide material, they remove those parts that that should not be developed into a MOF material. The approach is also described as bottom-up approach in [Metal organic framework top-down and bottom-up patterning techniques <http://dx.doi.org/10.1039/D0DT02207A>]. The process can be completely dry depending on how the ZnO material is patterned.

[RESPONSE] The approach by Ameloot et al. used a pre-patterned substrate, whereas, in our study, the process started from a planar, pattern-free substrate. We change the text (on Page 1) to "...but their application in MOF patterning **has not been widely realized.[33,34]**"

[COMMENT] **In the approach suggested by the authors, the unconverted layer with a carbon passivation layer will remain in between their patterns. This is hardly mentioned in the paper, and neither of how this will affect the properties of the resulting device/structure.**

[RESPONSE] In the original manuscript, we have included the discussion of the remaining ZnO layer in the text regarding Fig. 3:

"A cross-section of a region encompassing both ZIF-8 and ZIF-8-free adjacent areas (**Fig. 3c**) was prepared by focused ion beam (FIB) and examined by TEM (**Fig. 3d** and **e**) to elucidate the structure after the 2mIm vapor treatment. In the square irradiated with 20 mC/cm², the ZnO film remains intact, confirming that its conversion to ZIF-8 is entirely suppressed by the e-beam irradiation. In contrast, the ZnO in the non-irradiated area is mostly consumed after the vapor treatment. The ZIF-8 and ZIF-8-free areas are also clearly distinguished in SEM-EDS (**Fig. S2**), corresponding to the C/N-rich and C/N-deficient areas, respectively."

We agree with the reviewer that further emphasis should be added and we have updated the discussion regarding Fig. 4c to make a clear statement of the passivation layer (on Page 6):

"The passivation is likely a result of the e-beam induced crosslinking that transforms the adsorbed 2mIm molecules into an oligomerized or polymeric coating, which inhibits the conversion of ZnO to ZIF-8 in the irradiated area, while the non-irradiated ZnO surface remains reactive during the 2mIm vapor treatment."

We also included the discussion regarding the effect of unconverted ZnO (on Page 6):

"In the current approach, the passivated ZnO remains in the areas that do not convert to ZIF. For certain applications, the presence of ZnO may not be undesirable. For instance, when patterned MOFs are used as diffraction grating sensors [25, 50, 51], the diffraction intensity depends on the refractive index difference in different regions of the pattern. The non-MOF part of the pattern can be either empty or filled with metal oxide [50] as long as there is a spatial contrast in refractive index."

[COMMENT] **In addition, similar approaches are also described using X-ray or e-beam lithography in the papers:**

Top-down patterning of Zeolitic Imidazolate Framework composite thin films by deep X-ray lithography <http://dx.doi.org/10.1039/C2CC33292B>

Direct X-ray and electron-beam lithography of halogenated zeolitic imidazolate frameworks <https://doi.org/10.1038/s41563-020-00827-x>

The three latter works should be mentioned and dealt with in the current paper.

[RESPONSE] We have updated the introduction (on Page 1), now including the references suggested:

“Recently, it has been demonstrated that amorphization of ZIFs, under X-ray and e-beam irradiations, enables a selective removal of the irradiated or non-irradiated regions of the ZIFs in a liquid phase developing step. Deep X-ray lithography was also utilized to pattern ZIF films by selectively crosslinking a sol-gel bottom layer.[29]”

[COMMENT] Additional thoughts:

Page 5 line 123: The authors show that the amount of ZIF-8 formed is linearly dependent on the exposure time. 15 min = 50 nm and 60 min = 200 nm. The way I interpret this is that the conversion process is not complete in the first case, and may still not be so after 1 h. Primarily: When will it saturate? Where is the excess ZnO stored before conversion, and how does this affect the properties?

[RESPONSE] As demonstrated in the previous reports (Stassen et al., Nat. Mater. 2015, 15, 304-310 and Cruz et al., Chem. Mater. 2019, 31, 9462–9471), the oxide-to-MOF thickness expansion in our study should be similarly lower than ca. 16.8X (expected for ideal, single-crystal ZnO) due to the low-density ALD ZnO deposited under our condition. Considering around 15 nm of ZnO film in our study, ca. 200 nm of ZIF-8 film observed after 60 min of reaction time is regarded near a complete conversion with little residual ZnO left at the bottom.

The conversion of ZnO to ZIF-8 starts from ZIF-8 nucleation on the top surface. As the reaction proceeds, the ZIF-8 layer becomes thicker at the cost of the ZnO layer at the bottom. (Chem. Mater. 2019, 31, 9462–9471) The remaining ZnO serves as a native adhesion layer to ensure a good bond between the formed ZIF-8 and the substrate. Otherwise, an extra adhesion layer (e.g., TiO₂) underneath the ZnO may be necessary to ensure the uniformity of the resulting ZIF-8 film. (Nat. Mater. 2016, 15, 304-310).

In the revised manuscript, we have included additional discussion of Figure 3 to address the questions of the reviewer:

“A cross-section of a region encompassing both ZIF-8 and ZIF-8-free adjacent areas (**Fig. 3c**) was prepared by focused ion beam (FIB) and examined by TEM (**Fig. 3d** and **e**) to elucidate the structure after the 2mIm vapor treatment. In the square irradiated with 20 mC/cm², the ZnO film remains intact, confirming that its conversion to ZIF-8 is entirely suppressed by the e-beam irradiation. In contrast, the ZnO in the non-irradiated area is mostly consumed after the vapor treatment. In agreement with previous reports,[33,41] a thin unconverted layer of ZnO remains. It is located at the substrate-ZnO interface consistent with the proposed conversion of ZnO to ZIF-8 starting from the top of the film and propagating to the substrate-ZnO interface.[41] The presence of this thin unconverted layer could be beneficial for ensuring good adhesion of the ZIF-8 deposit to the substrate.[33] The ZIF-8 and ZIF-8-free areas are also clearly distinguished in SEM-EDS (**Fig. S2**), corresponding to the C/N-rich and C/N-deficient areas, respectively.”

The matter of the residual ZnO at the bottom and how it may affect ZIF deposit properties depends on the application of interest, as described in answer to comment #2 above.

REVIEWER COMMENTS

Reviewer #1 (Remarks to the Author):

I feel that the comments have been addressed in a satisfactory way.

Reviewer #3 (Remarks to the Author):

I do acknowledge that the authors have tried to amend the paper, however, it still contains shortcomings. I will try to elaborate:

In my original comment:

[COMMENT] Bottom-up patterning of MOF materials was also included in the original paper by Ameloot [ref. 36]. See its figure 5a and the description in the text that goes with it. Rather than to passivate parts of the oxide material, they remove those parts that that should not be developed into a MOF material. The approach is also described as bottom-up approach in [Metal organic framework top-down and bottom-up patterning techniques <http://dx.doi.org/10.1039/D0DT02207A>]. The process can be completely dry depending on how the ZnO material is patterned.

[RESPONSE] The approach by Ameloot et al. used a pre-patterned substrate, whereas, in our study, the process started from a planar, pattern-free substrate. We change the text (on Page 1) to "...but their application in MOF patterning has not been widely realized.[33,34]"

[R2] I do not understand the authors choice in answering here. Also the authors approach begin with a none-pre-patterned substrate before it is brought into the e-beam lithography technique. What is their clear benefit from the prior descriptions? Also patterning of the ZnO material can be done by dry processing. Why is the authors approach so novel when keeping in mind that an unreacted layer of ZnO is left behind under the decomposed imidazole?

Also: What do the authors try to gain by stating "but their application in MOF patterning has not been widely realized." When you can say the same about deposition of MOF materials as thin films in general?

This is one of the main aspects that prevents me from recommending this for Nature Communications. Yes, e-beam lithography is a cool thing, but I fail to recognize how it can be any better than other approaches that does not leave residues behind. If the authors had found an application where this residue took part in a device structure, then I would be more than convinced. Presently, not.

This also applies to how the current work relates to the works:

Top-down patterning of Zeolitic Imidazolate Framework composite thin films by deep X-ray lithography <http://dx.doi.org/10.1039/C2CC33292B>

Direct X-ray and electron-beam lithography of halogenated zeolitic imidazolate frameworks <https://doi.org/10.1038/s41563-020-00827-x>

The authors refers to one of these as information but does not discuss how it relates to their own findings. I am not convinced that the present work adds enough novelty and significance to reach Nature Communications. Other journals, yes.

2nd Response to Reviewers for
Solvent-Free Bottom-up Patterning of Zeolitic Imidazolate Frameworks

*Changes to the manuscript are highlighted in yellow

Reviewer #3 (Remarks to the Author):

[COMMENT] I do acknowledge that the authors have tried to amend the paper, however, it still contains shortcomings. I will try to elaborate:

In my original comment:

[comment] Bottom-up patterning of MOF materials was also included in the original paper by Ameloot [ref. 36]. See its figure 5a and the description in the text that goes with it. Rather than to passivate parts of the oxide material, they remove those parts that should not be developed into a MOF material. The approach is also described as bottom-up approach in [Metal organic framework top-down and bottom-up patterning techniques <http://dx.doi.org/10.1039/D0DT02207A>]. The process can be completely dry depending on how the ZnO material is patterned.

[response] The approach by Ameloot et al. used a pre-patterned substrate, whereas, in our study, the process started from a planar, pattern-free substrate. We change the text (on Page 1) to "...but their application in MOF patterning has not been widely realized.[33,34]"

[r2] I do not understand the authors choice in answering here. Also the authors approach begin with a none-pre-patterned substrate before it is brought into the e-beam lithography technique. What is their clear benefit from the prior descriptions? Also patterning of the ZnO material can be done by dry processing.

[RESPONSE] The work by Ameloot and co-workers (original ref. 36, current ref. 33) demonstrated patterning of ZIF-8 (Figure 5a in ref .33) using a lift-off patterning technique. A silicon wafer was covered with positive tone resist, exposed to UV through a mask and then developed in an aqueous alkaline solution (AZ developer). ZnO was then deposited by sputtering and then ZIF-8 was formed by vapor treatment. The photoresist was then lift-off by sonication in acetone. This process, although it has bottom-up elements and some steps are solvent-free, is not a solvent-free bottom-up patterning process.

To address the reviewer's comment, we modified the text as follows:

"Yet, solvent-free approaches for patterning are currently at the forefront of technological needs due to their great potential in improving wafer processing efficiency and patterning quality at reduced material and energy cost.[32] Although solvent-free MOF deposition steps have been incorporated in lift-off patterning,[33,34] solvent-free bottom-up MOF patterning has not yet been demonstrated."

[COMMENT] Why is the authors approach so novel when keeping in mind that an unreacted layer of ZnO is left behind under the decomposed imidazole?

[RESPONSE] Our approach is novel because solvent-free bottom-up ZIF processing has not been reported. The unconverted ZnO is not necessarily undesirable, as has been discussed already in the manuscript:

"In the current approach, the passivated ZnO remains in the areas that do not convert to ZIF. For certain applications, the presence of ZnO may not be undesirable. For instance, when patterned MOFs are used as diffraction grating sensors [25, 50, 51], the diffraction intensity depends on the refractive index difference

in different regions of the pattern. The non-MOF part of the pattern can be either empty or filled with metal oxide [50] as long as there is a spatial contrast in refractive index.”

[COMMENT] Also: What do the authors try to gain by stating “but their application in MOF patterning has not been widely realized.” When you can say the same about deposition of MOF materials as thin films in general?

[RESPONSE] As stated above, we have now removed this statement and replaced it with a clearer (we hope) statement.

[COMMENT] This is one of the main aspects that prevents me from recommending this for Nature Communications. Yes, e-beam lithography is a cool thing, but I fail to recognize how it can be any better than other approaches that does not leave residues behind. If the authors had found an application where this residue took part in a device structure, then I would be more than convinced. Presently, not.

[RESPONSE] As stated above, the presence of ZnO in the patterned structure may not be undesirable depending on the specific application. As the reviewer stated in the original review: “*Coatings of MOF type of materials, and patterning of such, is an emerging field that is still lacking a practical application*”. We are currently working to demonstrate applications of the discovery reported here.

[COMMENT] This also applies to how the current work relates to the works:

Top-down patterning of Zeolitic Imidazolate Framework composite thin films by deep X-ray lithography <http://dx.doi.org/10.1039/C2CC33292B>

Direct X-ray and electron-beam lithography of halogenated zeolitic imidazolate frameworks <https://doi.org/10.1038/s41563-020-00827-x>

The authors refers to one of these as information but does not discuss how it relates to their own findings. I am not convinced that the present work adds enough novelty and significance to reach Nature Communications. Other journals, yes.

[RESPONSE] To further highlight the differences between these works and ours, we also added the following text:

“...Moreover, top-down patterning methods [25-27, 29] rely on irradiation induced solubility change and subsequent removal of materials by dissolution. Yet, solvent-free approaches for patterning are currently at the forefront of technological needs due to their great potential in improving wafer processing efficiency and patterning quality at reduced material and energy cost.[32].”